# Progression of Chronic Kidney Disease and All-Cause Mortality in Patients with Tricuspid Regurgitation

**DOI:** 10.3390/diseases10010016

**Published:** 2022-03-16

**Authors:** Fabian Schipmann, Marwin Bannehr, Valentin Hähnel, Victoria Dworok, Jonathan Nübel, Christoph Edlinger, Michael Lichtenauer, Michael Haase, Michael Zänker, Christian Butter, Anja Haase-Fielitz

**Affiliations:** 1Department of Cardiology, Heart Center Brandenburg Bernau & Faculty of Health Sciences (FGW) Brandenburg, Brandenburg Medical School (MHB) Theodor Fontane, Ladeburger Straße 17, 16321 Bernau bei Berlin, Germany; fabian.schipmann@immanuelalbertinen.de (F.S.); marwin.bannehr@immanuelalbertinen.de (M.B.); valentin.haehner@immanuelalbertinen.de (V.H.); victoria.dworok@immanuelalbertinen.de (V.D.); jonathan.nuebel@immanuelalbertinen.de (J.N.); christophroland.edlinger@immanuelalbertinen.de (C.E.); christian.butter@immanuelalbertinen.de (C.B.); 2Clinic of Internal Medicine II, Department of Cardiology, Paracelsus Medical University of Salzburg, 5020 Salzburg, Austria; m.lichtenauer@salk.at; 3Medical Faculty, Otto von Guericke University Magdeburg, 39120 Magdeburg, Germany; michael.haase@med.ovgu.de; 4Department of Internal Medicine, Heart Center Brandenburg Bernau, Brandenburg Medical School (MHB) Theodor Fontane, 16321 Bernau bei Berlin, Germany; m.zaenker@immanuel.de; 5Institute of Social Medicine and Health System Research, Otto von Guericke University Magdeburg, 39120 Magdeburg, Germany

**Keywords:** tricuspid regurgitation, long-term mortality, chronic kidney disease, CKD progression, cardio-renal

## Abstract

Aim: The impact of chronic kidney disease (CKD) on patient-related outcomes in patients with tricuspid regurgitation (TR) is well known. However, the impact of the progression of CKD in patients with TR and potentially modifiable risk factors of progressing CKD is unknown. Methods: 444 consecutive adult patients with TR and CKD stage 1–4 admitted in an inpatient setting between January 2010 and December 2017 were included. During a median follow-up of two years, eGFR and survival status were collected. Independent risk factors for CKD progression and all-cause mortality were determined. Patient survival statuses were grouped according to different combinations of the presence or absence of CKD progression and the TR grade. Results: Progression of CKD (OR 2.38 (95% confidence interval 1.30–4.35), *p* = 0.005), the grade of TR (OR 2.38 (1.41–4.00), *p* = 0.001) and mitral regurgitation (OR 1.72 (1.20–2.46), *p* = 0.003) were independent risk factors for all-cause mortality. Haemoglobin at admission (OR 0.80 (0.65–0.99), *p* = 0.043) and the presence of type 2 diabetes (OR 1.67 (1.02–2.73), *p* = 0.042) were independent risk factors for CKD progression. The combination of the status of CKD progression and the TR grade showed a stepwise pattern for all-cause mortality (*p* < 0.001). Patients with CKD progression and TR grade 1 had comparable all-cause mortality with patients without CKD progression but with TR grade 2 or 3. Even in patients with TR grade 1, the risk for all-cause mortality doubled if CKD progression occurred (OR 2.49 (95% CI 1.38–4.47), *p* = 0.002). Conclusion: CKD progression appears to be a risk factor for all-cause mortality in patients with TR. Anaemia and diabetes are potential modifiers of CKD progression.

## 1. Introduction

Tricuspid regurgitation (TR) is the least reported of all the valve diseases but clearly associated with increased mortality [1,2,3,4]. Data on the prevalence of TR is relatively sparse and depends on the clinical setting. In the Framingham Heart Study, 82.0% of men and 85.7% of women showed signs of TR. Tricuspid regurgitation was thus the second most common valve disease after mitral valve regurgitation [5]. Topilsky et al. calculated the overall prevalence of at least moderate TR for the American population at 0.55%. Above the age of 75 years there is a significant increase to 3.96% [6].

For the treatment of patients with TR, only diuretics are currently used as a conservative therapeutic option for volume reduction. Surgical treatment has not shown promising results and is primarily recommended in patients undergoing open-heart surgery for mitral or aortic valve disease. Transcatheter solutions may be a new therapeutic approach; however, the devices are still in development and outcomes in terms of symptom relief and survival are largely unclear.

As therapeutic options are limited in this patient cohort, the identification of modifiable risk factors of TR is urgently needed. Known risk factors for adverse outcomes in patients with TR comprise reduced left and right ventricular ejection fraction, non-tricuspid valve disease, chronic obstructive pulmonary disease, and stable chronic kidney disease (CKD) [5,6,7,8,9,10]. However, such known risk factors do not fully explain the high mortality rate of patients with TR and are not modifiable per se. Along this line, the prognostic impact of CKD progression in patients with TR may have been underestimated. Kidney impairment causes the accumulation of uraemic toxins and pronounced changes in fluid and electrolytes status, contributing to increased morbidity and mortality in cardiac patients [11,12,13]. In patients with CKD, volume overload and uraemic toxins are involved in aortic calcification and likely valve impairment, due to systemic inflammation and dysregulated parathyroid hormone axis [13]. Progressive fluid overload with diastolic dysfunction could increase pulmonary capillary wedge pressure, which eventually may result in TR and further right heart volume overload [14]. Previous work in patients with decompensated heart failure has suggested that venous congestion is one of the most important haemodynamic factors driving renal dysfunction [15]. However, metabolic acidosis helped to prevent vascular calcification in vitro and in vivo [16,17,18] and may have protective effects in patients with TR.

An overview of the literature regarding “tricuspid regurgitation” and “chronic kidney disease” and “mortality” is shown in Appendix A. However, the progression of CKD has not yet been considered as a potentially modifiable risk factor for patients with TR. The aim of this study was to evaluate the potential impact of the progression of CKD over time on all-cause mortality in patients with TR.

## 2. Materials and Methods

In this single-centre, retrospective observational cohort study, we analysed data from 444 consecutive adult patients with TR, including the evaluation of renal function over time and the complete survival status. Patients were admitted to the Department of Cardiology, Heart Centre Brandenburg Bernau of the Brandenburg Medical School, from January 2010 to December 2017. Patients with TR were identified using transthoracic echocardiograms performed at our centre. Patients with CKD stage 5 and no renal function measurement during the follow-up were excluded. The patient flow through the study is shown in Figure 1. The STROBE reporting recommendations were used (Appendix A). The study was approved by the local Ethics Committee of the State of Brandenburg (AS 155(bB)/2017). The Ethics Committee waived written informed consent because the data analysed were collected as part of routine diagnosis and treatment including patient follow-up.

### 2.1. Data Collection

Demographic data, comorbidities, type of admission, type of diuretics, echocardiographic and laboratory parameters were obtained from the patient’s medical record, from admission through to hospital discharge. Routinely, at hospital admission, patients provided the type of outpatient specialist care they were under. Pulmonary hypertension was defined according to ESC criteria [19]. Echocardiographic examinations were performed by trained physicians using GE-Vingmed, Vivid 7 and E9, Horten Norway. Complete 2-dimensional echocardiograms included pulsed-wave, continuous-wave, and colour Doppler imaging. Standard echocardiographic measurements for the evaluation of the left atrial (LA) size and the left ventricular (LV) geometry and function were performed, according to the guidelines of the American Society of Echocardiography [20]. The determination of TR was conducted in line with current recommendations and guidelines, using the following criteria for severe TR: dilated annulus with no valve coaptation or flail leaflet, large central jet > 50% of the right atrial, vena contracta (VC) width > 7 mm, proximal isovelocity surface area (PISA) radius > 9 mm and an effective regurgitant orifice area > 0.40 cm^2^ at Nyquist 30–40 cm/s, dense triangular continuous wave jet or sine wave pattern and dilated RV. The PISA was obtained from an apical four chamber zoom view. Tricuspid regurgitation was graded, according to the most frequently used classification, into three grades: mild (TR 1), moderate (TR 2) and severe (TR 3) [21,22]. The laboratory admission value, the highest or lowest value and the value closest to the patient hospital discharge were obtained. Renal function parameters included serum creatinine, blood urea nitrogen (BUN), BUN/creatinine ratio and the estimated glomerular filtration rate (eGFR). Other routine laboratory values at hospital admission were NT-proBNP, Troponin I, HbA1c, haemoglobin, alanine transaminase (ALT), leukocytes, total cholesterol, sodium, and potassium.

### 2.2. Follow-Up

The follow-up was performed in July 2019. The patients were scheduled for a cardiology outpatient clinic visit at three months after the index hospital stay and then on a yearly basis, if not clinically needed otherwise. The estimated glomerular filtration rate, survival status and the date of death were collected from the outpatient medical record and the primary care physician.

### 2.3. Study Endpoint and Definitions

The primary endpoint was all cause-mortality. We defined CKD progression as non-reversible deterioration in the CKD stage during the follow-up, using the KDIGO criteria [23]. For example, a change from CKD stage 3 to CKD stage 4 was considered as CKD progression. The baseline eGFR was defined as the eGFR obtained after cardiac re-compensation during the index hospital stay. In a sensitivity analysis for all-cause mortality, CKD progression was defined as a sustained decline in the eGFR > 5 mL/min/1.73 m^2^ per year [23]. The stage of CKD was assessed using the admission and follow-up eGFR values. The estimated GFR was calculated using the serum creatinine-based CKD-EPI equation [24].

### 2.4. Statistical Analysis

Categorical variables were analysed with Fisher’s exact or a chi-square test, and continuous variables with a nonparametric Mann–Whitney U or Kruskal–Wallis test. Data are presented as median with 25th to 75th percentile or mean with the standard deviation for continuous variables and as a proportion for the categorical variables. The effects of a combination of different TR grades and CKD progression on all-cause mortality were analysed. Missing data are reported in Tables and Figures footnotes. To evaluate the impact of CKD progression on all-cause mortality, we performed multivariable logistic regression analysis, including variables with *p*-value < 0.1 or those considered to be clinically relevant. The following variables were included in the multivariable model for all-cause mortality: type of admission, CKD progression (stage-based or, in a sensitivity analysis: eGFR loss-based) during the follow-up, TR grade, MR grade, haemoglobin and NT-proBNP at admission, COPD, gender, type 2 diabetes, atrial fibrillation, and age. The following variables were included in the multivariable model for CKD progression: eGFR at admission, haemoglobin at admission, type 2 diabetes, LVEF (%), COPD, coronary artery disease, and age. The results are presented as odds ratios (OR) with 95% confidence intervals (CI). A two-sided *p* value of less than 0.05 was considered statistically significant. SPSS 26 (SPSS Inc., Chicago, IL, USA, 2019) was used. The dataset is available from the authors upon reasonable request.

## 3. Results

### 3.1. Patient Characteristics

Of the 444 patients included none of the patients died during the index hospital stay. During the follow-up (median 2 (1–3) years), 101 patients (22.7%) died. One- and three-year mortality rates were 4.9% and 50.7%, respectively. Demographic data, comorbidities, type of admission, diuretics, laboratory, and echocardiographic parameters according to mortality status are presented in Table 1. The patients who died during the follow-up were similar regarding demographic variables, NYHA class, and the proportions of patients presenting with arterial hypertension and dyslipidaemia compared to surviving patients. One hundred and thirty-two patients (29.7%) presented with CKD progression. The patients who died during the follow-up more frequently presented with CKD progression (44.6%) compared to patients who survived (25.4%), (odds ratio 2.37 (95% CI 1.49–3.75), *p* < 0.001) (Table 1). The median loss of eGFR over time was −15.3 mL/min (25th–75th percentile −27.5 to −4.3) in patients who died compared to −9.9 mL/min (25th–75th percentile −20.0 to −2.1) in those who survived (*p* = 0.023). Also, the patients who died presented with a higher grade of right atrial (RA), left atrial (LA) and right ventricular (RV) dilatation (all *p* < 0.001) compared to those who survived. A total of 16 out of 444 patients (3.6%) were admitted with Torasemide at a dose of >50 mg per day. The patients who died received a twice as high dose of Torasemide at hospital admission compared to the patients who survived; 20 (10–30) mg vs. 10 (10–20) mg, *p* < 0.001 (Table 1). Tolvaptan was not used in any patient to manage systemic congestion.

### 3.2. Independent Risk Factors for All-Cause Mortality

Progression of CKD during the follow-up (OR 2.38 (95% CI 1.30–4.35), *p* = 0.005), the TR grade and MR grade during the index hospital stay were independent risk factors for all-cause mortality (Table 2). The results remained essentially unchanged if the eGFR loss > 5 mL/min/1.73 m^2^ per year criterion instead of the CKD stage-based definition for CKD progression was used.

### 3.3. Progression of Chronic Kidney Disease

The patients with or without CKD progression were similar regarding age, gender, major cardiovascular disease, and echocardiographic parameters, as well as the grade of tricuspid and mitral valve regurgitation (Table 3). The patients with CKD progression presented more frequently with diabetes (43.9%) compared to those without CKD progression (30.5%) (OR 1.79 (95% CI 1.18–2.72), *p* = 0.007). Moreover, the patients with CKD progression showed higher admission eGFR compared to patients without progression of CKD (Table 3). At hospital admission, the dose of Torasemide according to the grade of TR did not differ in patients with and without CKD progression (*p* > 0.15). In the patients with CKD progression, more patients died compared to patients without progression (34.1% vs. 18.0%, OR 2.37 (95% CI 1.49–3.75), *p* < 0.001). In addition, the length of stay in hospital was similar in patients with CKD progression (8 (5–14) days) compared to those without CKD progression (8 (4–12) days) (*p* = 0.107).

### 3.4. Independent Risk Factors for CKD Progression

In multivariable logistic regression analysis, admission haemoglobin (OR 0.80 (95% CI 0.65–0.99), *p* = 0.043) and the presence of type 2 diabetes (OR 1.67 (95% CI 1.02–2.73), *p* = 0.042) were identified as independent risk factors for CKD progression (Table 4). All-cause mortality was analysed according to the combination of the TR grade and the status of CKD progression. Most patients presented with TR stage 1 and no CKD progression (235/444, 52.9%) and 40/444 (9.0%) had both TR stage 2 or 3 and CKD progression. The combination of the stage of TR and the status of CKD progression showed a stepwise pattern for all-cause mortality (Figure 2). The patients with TR stage 1 and CKD progression had similar all-cause mortality to the patients with TR stage 2 or 3 but no CKD progression. In patients with TR stage 1, the risk for all-cause mortality doubled if CKD progression occurred (OR 2.49 (95% CI 1.38–4.47), *p* = 0.002).

## 4. Discussion

In 444 consecutive patients with TR, we characterised patients according to survival status and assessed the prognostic impact of progressing CKD and non-kidney related factors. In a typical cohort of patients with TR, we found the progression of CKD, the grade of TR and MR to be independent risk factors for all-cause mortality. All-cause mortality doubled in patients with CKD progression compared to patients without progression. Furthermore, we aimed to identify independent risk factors for CKD progression, affecting about one third of the patients. Lower admission haemoglobin concentration and the presence of diabetes were identified as independent risk factors for CKD progression. For scoping the meaning of CKD progression, we evaluated all-cause mortality according to different combinations of TR grade and status of CKD progression. The patients with TR stage 1 and CKD progression had similar all-cause mortality to the patients with TR stage 2 or 3 but no CKD progression. The mortality rate reported by Mehr et al. [25] was 20% and similar to the rate we observed. However, the patients in this previous observational cohort study received TR-related intervention and the observation period was only one year [25]. One-year mortality in the patients with severe TR was 24% [25] and the five-year mortality rate of patients with TR complicating heart failure, with reduced ejection fraction, was 57% [26]. The risk factors for long-term mortality identified in other studies comprise impaired LV function, several echocardiographic parameters such as TAPSE < 16 mm, COPD, anaemia, and CKD [4,9,27]. Although it is known that in patients with TR, the severity of CKD is closely related to mortality [28], the progression of CKD and its prognostic impact has not yet been investigated. Of note, unlike CKD presence at hospital admission, progression of CKD is a potentially modifiable factor for mortality. In this study, CKD progression was a strong risk factor for all-cause mortality. Moreover, both lower haemoglobin and diabetes constitute potentially modifiable risk factors for CKD progression, especially regarding the treatment of renal anaemia and diabetic nephropathy. Pathophysiological considerations and the known effects of CKD on the course and prognosis of patients with valvular disease render the results of this study to be plausible. Reduced glomerular filtration rates in patients with CKD causes the systemic accumulation of uraemic toxins, which has been correlated with cardiac disease progression and increased morbidity [29,30]. Specifically, uraemic toxins and excessive changes in circulating blood volume led to the activation of the renin-angiotensin system, increased oxidative stress, inflammation, and fibrosis [31,32], contributing to progressive sclerosis and calcification of the valves and valvular annuli. Valve fibrosis and calcification may be aggravated by hypoxia and CKD-related fluid overload. Indeed, anaemia was a risk factor for CKD progression and all-cause mortality in this patient cohort. In this study cohort, especially in the patients with mild TR, CKD progression was prognostically relevant. If confirmed, this finding may be relevant for early kidney care in patients with TR. Based on the study findings, the evaluation of kidney follow-up may be advisable, especially in patients with mild TR. In patients with TR, specific focus may be put on diabetes care and anaemia management. Comprehensive clinical, laboratory and echocardiographic databases were merged. The patient cohort was relatively large and included patients with functional/secondary TR. The staging of tricuspid regurgitation as a covariable was based on established guideline recommendations [20]. Long-term follow-up was provided, but the data recorded were limited to survival status and renal function. Serum creatinine-based GFR estimation is limited by muscle mass and protein intake and may apply to this patient cohort, given that nutritional impairment is common in patients with TR [33]. Unfortunately, we were not able to stage CKD according to proteinuria criteria and to provide serum creatinine-independent kidney function parameters, such as serum cystatin C. Future studies may clarify the role of kidney care, especially in patients with diabetes and anaemia, and monitoring of additional renal function parameters such as serum cystatin C and Dickkopf-3 in patients with TR [34]. In addition, the prognostic impact of the treatment of renal anaemia and diabetic nephropathy in patients with TR needs to be evaluated.

## 5. Conclusions

Based on the study findings, beyond the TR and MR grade, CKD progression is an independent risk factor for all-cause mortality in patients with TR. Anaemia and diabetes are potentially modifiable risk factors of CKD progression in patients with TR. Patients with TR stage 1 must be especially considered for kidney follow-up. The study findings need to be validated in prospective studies.

## Figures and Tables

**Figure 1 diseases-10-00016-f001:**
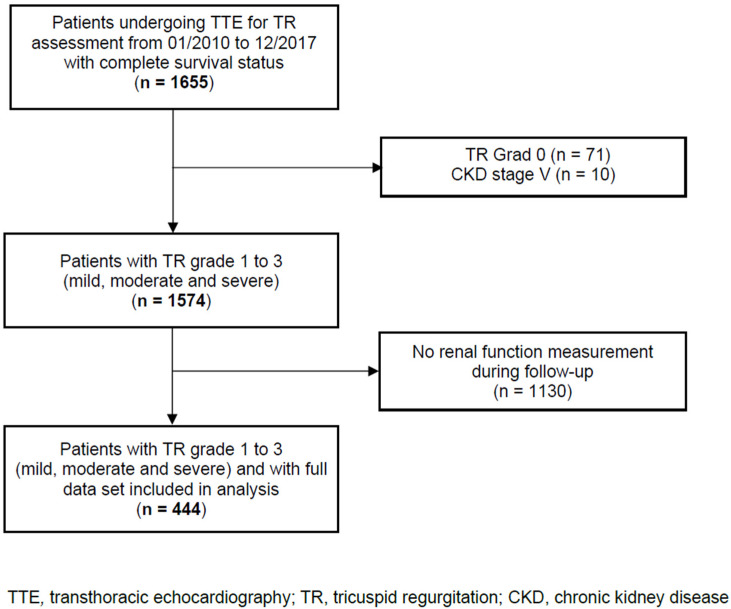
Patient flow through the study.

**Figure 2 diseases-10-00016-f002:**
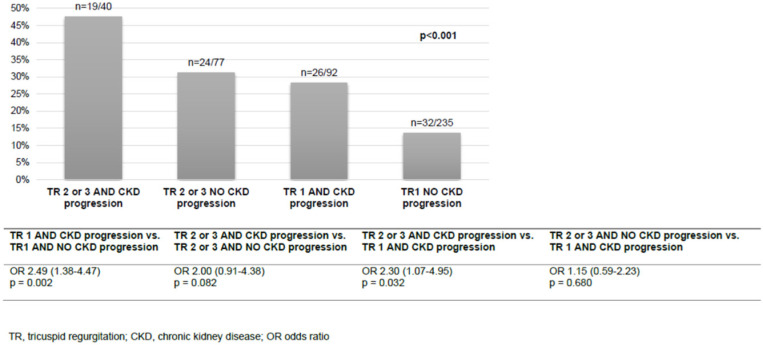
All-cause mortality according to the grade of TR and CKD progression.

**Table 1 diseases-10-00016-t001:** Patient characteristics according to all-cause mortality status.

	Died (*n* = 101)	Survived (*n* = 343)	*p*-Value
**Demographics**			
Age (years)	77 (73–82)	78 (72–82)	0.698
Female, *n* (%)	41 (40.6%)	161 (46.9%)	0.260
Body mass index (kg/m^2^)	26.5 (24.0–30.5)	26.7 (24.0–30.4)	0.878
**Type of admission**			
Elective case, *n* (%)	37 (37.0%)	172 (52.3%)	0.012
Transfer from other hospital, *n* (%)	19 (19.0%)	61 (18.5%)	
Emergency case, *n* (%)	44 (44.0%)	96 (29.2%)	
**Comorbidities**			
NYHA class, *n* (%)			
I + II	53 (52.5%)	184 (53.6%)	0.210
III	47 (46.5%)	151 (44.0%)	
IV	1 (1.0%)	8 (2.3%)	
Chronic kidney disease, *n* (%)			
I–II	37 (36.6%)	164 (46.9%)	0.029
III–IV	64 (63.4%)	179 (52.2%)	
Chronic kidney disease progression, *n* (%)	45 (44.6%)	87 (25.4%)	<0.001
Coronary artery disease, *n* (%)	47 (46.5%)	133 (38.8%)	0.169
Atrial fibrillation, *n* (%)	47 (46.5%)	108 (31.5%)	0.006
Arterial hypertension, *n* (%)	93 (92.1%)	325 (94.8%)	0.259
Pulmonary hypertension, *n* (%)	37 (36.6%)	99 (28.9%)	0.016
Dilatative cardiomyopathy, *n* (%)	11 (10.9%)	27 (7.9%)	0.345
Dyslipidaemia, *n* (%)	75 (74.3%)	243 (70.9%)	0.529
Type 2 diabetes, *n* (%)	46 (45.6%)	107 (31.2%)	0.008
Chronic obstructive pulmonary disease, *n* (%)	23 (22.8%)	43 (12.5%)	0.011
Cardiac device, *n* (%)	26 (25.7%)	77 (22.5%)	0.491
Tricuspid regurgitation, *n* (%)			
Mild TR	58 (57.4%)	268 (78.1%)	<0.001
Moderate TR	28 (27.7%)	64 (18.7%)	
Severe TR	15 (14.9%)	11 (3.2%)	
Mitral regurgitation, *n* (%)			
Mild MR	38 (37.6%)	177 (51.6%)	<0.001
Moderate MR	30 (29.7%)	93 (27.1%)	
Severe MR	32 (31.7%)	53 (15.5%)	
Aortic regurgitation, *n* (%)	35 (34.7%)	129 (37.6%)	0.758
**Laboratory results at admission**			
Estimated glomerular filtration rate (mL/min/1.73)	58.4 (42.8–74.4)	68.0 (54.8–83.9)	<0.001
Blood urea nitrogen/Creatinine ratio	22.4 (17.9–28.0)	20.7 (17.2–24.3)	0.032
NT-proBNP (pg/mL)	3791 (1994–7233)	2118 (731–4557)	<0.001
Troponin I (µg/L)	0.05 (0.03–0.08)	0.03 (0.01–0.05)	0.052
Haemoglobin (mmol/L)	7.6 (6.8–8.2)	7.9 (7.2–8.6)	0.001
Haematocrit (%)	37.0 (32.5–39.5)	38.0 (35.0–42.0)	<0.001
HbA1c (%)	6.0 (5.7–6.7)	5.9 (5.5–6.6)	0.170
Leukocytes (Gpt/L)	7.9 (6.9–10.1)	7.6 (6.4–9.4)	0.433
ALT (µkat/L)	0.35 (0.28–0.45)	0.37 (0.27–0.53)	0.171
Cholesterol (mmol/L)	4.2 (3.4–5.2)	4.8 (3.9–5.7)	0.016
Potassium (mmol/L)	4.4 (4.1–4.8)	4.4 (4.1–4.7)	0.340
Sodium (mmol/L)	138.5 (136.2–140.4)	139.0 (136.5–140.6)	0.614
**Diuretics at admission**			
Torasemide, *n* (%)	69 (68.3%)	231 (67.4%)	0.855
Torasemide dose, mg	20 (10–30)	10 (10–20)	0.001
Torasemide dose, mg			
Mild TR	20 (10–26)	10 (5–20)	0.006
Moderate TR	23 (10–40)	10 (10–20)	0.053
Severe TR	10 (10–50)	20 (10–20)	0.555
Aldosterone antagonists, *n* (%)	38 (37.6%)	108 (31.6%)	0.256
Hydrochlorothiazide, *n* (%)	17 (16.8%)	59 (17.2%)	0.931
**Echocardiographic Parameters**			
TAPSE (mm)	18.0 (15.0–22.0)	19.0 (16.0–23.0)	0.097
TR Velocity max (m/s)	3.2 (2.8–3.7)	3.0 (2.6–3.4)	0.009
LVEF (%)	48.0 (32.0–60.0)	50.0 (35.0–60.0)	0.147
LA Dilatation, *n* (%)			
None	13 (12.9%)	88 (25.7%)	<0.001
Mild	30 (29.7%)	135 (39.4%)	
Moderate	42 (41.6%)	90 (26.2%)	
Severe	15 (14.9%)	30 (8.8%)	
RA Dilatation, *n* (%)			
None	41 (40.6%)	196 (57.1%)	<0.001
Mild	27 (26.7%)	102 (29.7%)	
Moderate	23 (22.8%)	31 (9.0%)	
Severe	10 (9.9%)	13 (3.8%)	
RV Dilatation, *n* (%)			
None	63 (62.4%)	287 (83.7%)	<0.001
Mild	20 (19.8%)	32 (9.3%)	
Moderate	12 (11.9%)	19 (5.5%)	
Severe	6 (5.9%)	5 (1.5%)	

TR, tricuspid regurgitation; MR, mitral regurgitation. Missing data for each variable: <1%. HbA1c, glycated haemoglobin; ALT, alanine aminotransferase; TR, tricuspid regurgitation. Missing values for each variable: <5% (except for Troponin I: *n* = 200; HbA1c: *n* = 122; Leukocytes: *n* = 79; Cholesterol: *n* = 129). TAPSE, tricuspid annular plain systolic excursion; LVEF, left ventricular ejection fraction; LA, left atrial; RA, right atrial; RV, right ventricular. Missing data for each variable: <1%.

**Table 2 diseases-10-00016-t002:** Multivariable logistic regression analysis for all-cause mortality.

All-Cause Mortality	Odds Ratio	95% CI		*p*-Value
**TR Grade**	2.38	1.41	4.00	0.001
MR Grade	1.72	1.20	2.46	0.003
CKD progression	2.38	1.30	4.35	0.005
COPD	1.70	0.81	3.57	0.162
Haemoglobin at admission	0.83	0.63	1.08	0.166
NT-proBNP at admission	1.00	1.00	1.00	0.172
Type 2 diabetes	1.46	0.79	2.69	0.225
Gender	1.28	0.69	2.37	0.438
Type of admission	1.10	0.75	1.63	0.622
Age	1.00	0.97	1.04	0.710
Atrial fibrillation	1.09	0.57	2.07	0.792

CKD, chronic kidney disease; TR, tricuspid regurgitation; MR, mitral valve regurgitation; COPD, chronic obstructive pulmonary disease.

**Table 3 diseases-10-00016-t003:** Patient characteristics according to progression of chronic kidney disease.

	CKD Progression (*n* = 132)	No CKD Progression (*n* = 312)	*p*-Value
**Demographics**			
Age (years)	77 (70–83)	78 (73–82)	0.390
Female, *n* (%)	61 (46.2%)	141 (45.2%)	0.844
Body mass index (kg/m^2^)	26.9 (24.2–30.5)	26.6 (23.9–30.3)	0.515
**Type of admission**			
Elective case, *n* (%)	56 (42.4%)	153 (49.0%)	0.320
Transfer from other hospital, *n* (%)	28 (21.2%)	52 (16.7%)	
Emergency case, *n* (%)	45 (34.1%)	95 (30.5%)	
**Comorbidities**			
NYHA class, *n* (%)			
I + II	71 (53.8%)	166 (53.2%)	0.261
III	58 (43.9%)	140 (44.9%)	
IV	3 (2.3%)	6 (1.9%)	
Chronic kidney disease, *n* (%)			
I–II	90 (68.2%)	111 (34.5%)	<0.001
III–V	42 (31.8%)	201 (64.4%)	
Coronary artery disease, *n* (%)	45 (34.1%)	135 (43.3%)	0.068
Atrial fibrillation, *n* (%)	46 (34.9%)	109 (34.9%)	0.968
Arterial hypertension, *n* (%)	125 (94.7%)	293 (93.9%)	0.840
Pulmonary hypertension, *n* (%)	39 (29.6%)	97 (31.1%)	0.682
Dilatative cardiomyopathy, *n* (%)	9 (6.8%)	29 (9.3%)	0.389
Dyslipidaemia, *n* (%)	90 (68.2%)	228 (73.1%)	0.273
Type 2 diabetes, *n* (%)	58 (43.9%)	95 (30.5%)	0.007
Chronic obstructive pulmonary disease, *n* (%)	25 (18.9%)	41 (13.1%)	0.120
Cardiac device, *n* (%)	28 (21.2%)	75 (24.0%)	0.519
Tricuspid regurgitation, *n* (%)			
Mild TR	92 (69.7%)	234 (75.0%)	0.474
Moderate TR	32 (24.2%)	60 (19.2%)	
Severe TR	8 (6.1%)	18 (5.8%)	
Mitral regurgitation, *n* (%)			
Mild MR	61 (46.2%)	154 (49.4%)	0.818
Moderate MR	37 (28.0%)	86 (27.6%)	
Severe MR	26 (19.7%)	59 (18.9%)	
Aortic regurgitation, *n* (%)	52 (39.4%)	112 (35.9%)	0.615
**Laboratory results at admission**			
Estimated glomerular filtration rate (mL/min/1.73)	71.5 (54.1–83.8)	61.6 (50.5–82.4)	0.039
Blood urea nitrogen/Creatinine ratio	21.4 (18.2–26.0)	20.8 (17.1–24.8)	0.224
NT-proBNP (pg/mL)	2900 (1140–6175)	2560 (834–4701)	0.180
Troponin I (µg/L)	0.03 (0.01–0.05)	0.03 (0.01–0.06)	0.744
Haemoglobin (mmol/L)	7.7 (6.8–8.5)	7.9 (7.2–8.6)	0.087
Haematocrit (%)	38.0 (33.0–41.0)	38.0 (35.0–41.8)	0.121
HbA1c (%)	6.0 (5.6–6.7)	5.9 (5.5–6.6)	0.099
Leukocytes (Gpt/L)	8.0 (6.7–10.0)	7.5 (6.3–9.0)	0.043
ALT (µkat/L)	0.36 (0.28–0.50)	0.36 (0.27–0.52)	0.799
Cholesterol (mmol/L)	4.7 (3.7–5.4)	4.7 (3.8–5.8)	0.594
Potassium (mmol/L)	4.4 (4.0–4.7)	4.3 (4.1–4.7)	0.451
Sodium (mmol/L)	138.5 (136.3–140.3)	139.0 (136.5–140.7)	0.267
**Diuretics at admission**			
Torasemide, *n* (%)	94 (71.2%)	206 (66.0%)	0.286
Torasemide dose, mg	10 (10–20)	10 (10–20)	0.907
Torasemide dose, mg			
Mild TR	10 (6–20)	10 (5–20)	0.547
Moderate TR	10 (10–20)	20 (10–30)	0.183
Severe TR	15 (10–50)	10 (10–20)	0.823
Aldosterone antagonists, *n* (%)	45 (34.1%)	101 (32.4%)	0.725
Hydrochlorothiazide, *n* (%)	23 (17.4%)	53 (17.0%)	0.911
**Echocardiographic Parameters**			
TAPSE (mm)	19.0 (16.0–22.5)	18.0 (16.0–23.0)	0.942
TR Velocity max (m/s)	3.0 (2.7–3.3)	3.0 (2.6–3.4)	0.697
LVEF (%)	50.0 (30.0–60.0)	50.0 (38.0–60.0)	0.205
LA Dilatation, *n* (%)			
None	25 (18.9%)	76 (24.4%)	0.525
Mild	55 (41.7%)	110 (35.3%)	
Moderate	39 (29.6%)	93 (29.8%)	
Severe	13 (9.9%)	32 (10.3%)	
RA Dilatation, *n* (%)			
None	69 (52.3%)	168 (53.9%)	0.947
Mild	40 (30.3%)	89 (28.5%)	
Moderate	17 (12.9%)	37 (11.9%)	
Severe	6 (4.6%)	17 (5.5%)	
RV Dilatation, *n* (%)			
None	100 (75.8%)	250 (80.1%)	0.437
Mild	18 (13.6%)	34 (10.9%)	
Moderate	12 (9.1%)	19 (6.1%)	
Severe	2 (1.5%)	9 (2.9%)	

TR, tricuspid regurgitation; MR, mitral regurgitation. Missing data for each variable: <1%. CKD, chronic kidney disease; HbA1c, glycated haemoglobin; ALT, alanine aminotransferase; TR, tricuspid regurgitation. Missing values for each variable: <5% (except for Troponin I: *n* = 200; HbA1c: *n* = 122; Leukocytes: *n* = 79; Cholesterol: *n* = 129). TAPSE, tricuspid annular plain systolic excursion; TR Velocity max, maximal tricuspid regurgitation velocity; LVEF, left ventricular ejection fraction; LA, left atrial; RA, right atrial; RV, right ventricular. Missing values for each variable: <1%.

**Table 4 diseases-10-00016-t004:** Multivariable logistic regression analysis for CKD progression.

CKD Progression	Odds Ratio	95% CI		*p*-Value
Type 2 diabetes	1.67	1.02	2.73	0.042
Haemoglobin at admission	0.80	0.65	0.99	0.043
COPD	1.73	0.93	3.20	0.082
eGFR at admission	1.01	0.99	1.02	0.113
CAD	0.72	0.45	1.17	0.190
Age	0.99	0.96	1.01	0.314
LVEF	0.99	0.98	1.01	0.337

CKD, chronic kidney disease; eGFR, estimated glomerular filtration rate; LVEF, left ventricular ejection fraction; COPD, chronic obstructive pulmonary disease; CAD, coronary artery disease.

## Data Availability

Data will be provided upon reasonable request to the corresponding author.

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
