# Peer review of "Progression of Chronic Kidney Disease and All-Cause Mortality in Patients with Tricuspid Regurgitation"

_diseases, 2022, doi:10.3390/diseases10010016_

Round 1
Reviewer 1 Report
The authors provided a valuable data for the risk study of TR and CKD. Data analysis confirmed the promoting role of CKD in the occurrence and development of TR disease. It is very useful for cardiologists and nephrologists to take care of patients with TR and CKD.
Some minor points have to be improved:
1、The language has to be improved and The structure of the article needs to be more hierarchical, which is conducive to readers' reading. For example, in the introduction, give more information about the epidemiology of TR and possible effects of CKD in the pathogenesis of TR,which could be provided with two parts;
2、The ordinate of Figure 2 is not marked with parameters
3、For cholesterol in Table 1 and table 3, please indicate whether LDL-C or Tch
Author Response
Reviewer #1:
The authors provided a valuable data for the risk study of TR and CKD. Data analysis confirmed the promoting role of CKD in the occurrence and development of TR disease. It is very useful for cardiologists and nephrologists to take care of patients with TR and CKD.
Some minor points have to be improved:
1、The language has to be improved and the structure of the article needs to be more hierarchical, which is conducive to readers' reading. For example, in the introduction, give more information about the epidemiology of TR and possible effects of CKD in the pathogenesis of TR, which could be provided with two parts.
RESPONSE: We agree and ADDED a PARAGRAPH IN THE INTRODUCTION SECTION.
2、The ordinate of Figure 2 is not marked with parameters
RESPONSE: WE ADDED A TITLE TO THE ORDINATE.
3、For cholesterol in Table 1 and table 3, please indicate whether LDL-C or Tch
RESPONSE: THANK YOU FOR THIS VALUABLE COMMENT. We now indicate THROUGHOUT THE REVISED MANUSCRIPT, that we measured total cholesterol in our patients.
Reviewer 2 Report
This clinical study examines whether there is a link between progression of chronic kidney disease (CKD) and mortality of patients with tricuspid regurgitation (TR). A relatively large number of patients were included over a 7-year period to allow for a reasonable follow-up period. The data demonstrates that CKD progression does appear to be associated with increased TR mortality and hence should be assessed in patients with TR with a view to attempting to alleviate the CKD.
This is an interesting study that appears to have been well conducted. The study results are clear and comprehensive. The emphasis is put onto CKD but there also appears to be other risk factors that could be involved in the TR mortality, and these might deserve some additional emphasis! The manuscript has been well written.
Specific Comments.
Abstract:
- But presumably the impact of CKD on TR is only 'well known' to those that study it? What are the impacts?
- So, is progression of CKD measured by decreasing eGFR?
- How is CKD progression measured?
- What is meant by "Haemoglobin at admission'? Presumably all patients had a haemoglobin! Should this be 'a low haemoglobin' if anaemia associated with CKD progression?
Introduction:
- Indicate why therapeutic options are limited!
- Supplementary tables have not been provided!
Methods:
- Why is only eGFR mentioned in the Abstract when a detailed list of other measurements has been given in the Methods?
- Give more detail on how this eGFR is calculated!
Results:
- What is Torasemide?
Discussion:
Discussion is reasonable. There seems to be a number of risk factors linked to TR; these should perhaps be given as much emphasis as CKD. However, the authors do not give any indication of what can be done in these patients to alleviate the risk factors and whether this may provide an effective therapy.
Author Response
Reviewer #2:
This clinical study examines whether there is a link between progression of chronic kidney disease (CKD) and mortality of patients with tricuspid regurgitation (TR). A relatively large number of patients were included over a 7-year period to allow for a reasonable follow-up period. The data demonstrates that CKD progression does appear to be associated with increased TR mortality and hence should be assessed in patients with TR with a view to attempting to alleviate the CKD.
This is an interesting study that appears to have been well conducted. The study results are clear and comprehensive. The emphasis is put onto CKD but there also appears to be other risk factors that could be involved in the TR mortality, and these might deserve some additional emphasis! The manuscript has been well written.
Specific Comments.
Abstract:
But presumably the impact of CKD on TR is only 'well known' to those that study it? What are the impacts?
RESPONSE:
WE AGREE AND HAVE ADJUSTED PHRASING OF THIS SENTENCE OF THE ABSTRACT OF THE REVISED MANUSCRIPT.
So, is progression of CKD measured by decreasing eGFR? How is CKD progression measured?
RESPONSE:
THANK YOU FOR THIS COMMENT. WE HAVE INCLUDED THE definition OF CKD PROGRESSION UNDER THE HEADING ‘Study Endpoints AND DEFINITIONS’.
THE PARAGRAPH NOW READS:
“CKD progression was defined as a sustained decline in eGFR of more than > 5 mL/min/1.73m2 per year”
What is meant by "Haemoglobin at admission'? Presumably all patients had a haemoglobin!
RESPONSE:
haemoglobin was measured as routine laboratory parameter at hospital admission. Missing values for haemoglobin at hospital admission were < 5%.
we have made this clear in the methods section and in the footnote of table 1 in the revised manuscript.
Should this be 'a low haemoglobin' if anaemia associated with CKD progression?
RESPONSE:
THIS IS CORRECT. a low haemoglobin associated with CKD progression. IN TURN, A HIGH HAEMOGLOBIN AT HOSPITAL ADMISSION ASSOCIATED WITH A DECREASED ODDS RATIO FOR CKD PROGRESSION (TABLE 4).
Introduction: Indicate why therapeutic options are limited!
RESPONSE:
WE agree and added a section in the introduction section.
“For the treatment of patients with TR, only diuretics are currently used as a conservative therapeutic option for volume reduction. Surgical treatment has not shown promising results and is primarily recommended in patients undergoing open-heart surgery for mitral or aortic valve disease. Transcatheter solutions may thus be a new therapeutic approach, however devices are still in development and outcomes in terms of symptom relief and survival are largely unclear.”
Supplementary tables have not been provided!
RESPONSE: WE APPOLOGIZE; AND NOW PROVIDE SUPPLEMENTARY TABLES 1 anD 2.
Methods: Why is only eGFR mentioned in the Abstract when a detailed list of other measurements has been given in the Methods? Give more detail on how this eGFR is calculated!
RESPONSE:
we AGREE, HOWEVER DUE TO THE LIMITED WORD COUNT IN THE ABSTRACT WE SUGGEST FOCUSSING ON Egfr HERE. Estimated GFR was calculated using the serum creatinine-based CKD-EPI equation
(Levey AS, Stevens LA. Estimating GFR using the CKD Epidemiology Collaboration (CKD-EPI) creatinine equation: more accurate GFR estimates, lower CKD prevalence estimates, and better risk predictions. Am J Kidney Dis 2010;55:622-627).
Results: What is Torasemide?
RESPONSE: TORASEMIDE IS A LOOP DIURETIC AND ANTIHYPERTENSIVE DRUG.
Discussion: Discussion is reasonable. There seems to be a number of risk factors linked to TR; these should perhaps be given as much emphasis as CKD. However, the authors do not give any indication of what can be done in these patients to alleviate the risk factors and whether this may provide an effective therapy
RESPONSE:
WE agree and changed the discuSsion acCORDINGLY:
“Future studies may clarify the role of kidney care especially in patients with diabetes and anaemia and monitoring of additional renal function parameters such as serum cystatin C and Dickkopf-3 in patients with TR (31).”